# Enhanced CRC Growth in Iron-Rich Environment, Facts and Speculations

**DOI:** 10.3390/ijms252212389

**Published:** 2024-11-19

**Authors:** Marcello Chieppa, Marianna Kashyrina, Alessandro Miraglia, Diana Vardanyan

**Affiliations:** 1Department of Experimental Medicine, University of Salento Centro Ecotekne, S.P.6, 73100 Lecce, Italy; marianna.kashyrina@unisalento.it (M.K.); diana.vardanyan@unisalento.it (D.V.); 2Institute of Science of Food Production, Unit of Lecce, C.N.R., 73100 Lecce, Italy; alessandro.miraglia@ispa.cnr.it

**Keywords:** iron metabolism, colorectal cancer, intestinal microbiota

## Abstract

The contribution of nutritional factors to disease development has been demonstrated for several chronic conditions including obesity, type 2 diabetes, metabolic syndrome, and about 30 percent of cancers. Nutrients include macronutrients and micronutrients, which are required in large and trace quantities, respectively. Macronutrients, which include protein, carbohydrates, and lipids, are mainly involved in energy production and biomolecule synthesis; micronutrients include vitamins and minerals, which are mainly involved in immune functions, enzymatic reactions, blood clotting, and gene transcription. Among the numerous micronutrients potentially involved in disease development, the present review will focus on iron and its relation to tumor development. Recent advances in the understanding of iron-related proteins accumulating in the tumor microenvironment shed light on the pivotal role of iron availability in sustaining pathological tumor hallmarks, including cell cycle regulation, angiogenesis, and metastasis.

## 1. Introduction

Iron is the second most abundant element on Earth. Its symbol is Fe, from the Latin word “Ferrum” and its atomic number is 26. Iron is a transition element that exists in two different oxidation states: +3 and +2. The +2 state is obtained by giving electrons, while the +3 state is obtained by donating electrons. Iron is a mineral that is naturally present in many foods that the body uses, mostly as an essential component of hemoglobin, the main protein that transfers oxygen from lungs to tissues [1]. Due to iron’s unique ability to accept or donate electrons, iron proteins are widely active in various cellular processes, including DNA synthesis, nucleic acid repair, mitochondrial respiration, cell proliferation, and apoptosis; they also play an important role in host defense and cellular signaling [2].

Iron is the most abundant trace metal in the human body, as its total content in a 70 kg healthy man is approximately 3–4 g (35–45 mg of iron per kilogram of body weight) compared to 2–3 g of zinc (the second most abundant) and 110 mg of copper (the third most abundant) [3]. Most iron is found in erythroid cells (>2 g) as heme in hemoglobin or in muscle myoglobin (~300 mg). Macrophages in the liver, spleen, and bone marrow retain a transient fraction of iron (~600 mg), while excess iron is stored in the liver parenchyma as ferritin (~1 g) [4]. Mammalian cells contain the following distribution of iron: roughly 16 mg/m^2^ in the mitochondria, 6 mg/m^2^ in the cytoplasm, 7 mg/m^2^ in the nucleus, and 16 mg/m^2^ in the lysosomes [5]. In a state of homeostasis, the human body maintains iron balance by regulating the daily intake and excretion of iron, with approximately 1 to 2 mg of iron being absorbed and eliminated each day [6].

There are two main sources of dietary iron: heme iron and non-heme (or inorganic) iron. The two forms of dietary iron are absorbed with different degrees of efficiency. The main sources of heme iron are the hemoglobin and myoglobin found in meat, seafood, and poultry. The absorption of heme iron is efficient and largely unaffected by other diet components. Plant based foods such as cereals and pasta, beans, and dark green leafy vegetables are the main sources of non-heme iron, which is highly insoluble; its bioavailability is influenced by many dietary components, such as ascorbic acid [7,8]. The ability of ascorbic acid to increase iron absorption may be related to its ability to reduce ferric to ferrous iron in the gut lumen. This may favor iron uptake by the divalent metal ion transporter 1 (DMT1). Furthermore, ascorbic acid may chelate soluble iron in the acidic pH environment of the stomach and preserve the soluble form in the increased pH environment of the duodenum. In 2023, A.K. von Siebenthal et al. demonstrated that ascorbic acid supplementation may increase iron absorption by 30% in iron deficient women, if administered in the morning. In the same study, coffee, polyphenols, and phytic acid reduced iron absorption even in the presence of ascorbic acid [9]. Nutritional derived components present in seeds and vegetables, (e.g., polyphenols, phytates) sequestrate non-heme iron and prevent its absorption [10,11].

Both heme and non-heme iron are absorbed in the duodenum and proximal jejunum. Although the dietary intake of iron is minimal, the regulation of intestinal iron absorption is crucial due to the absence of a physiological pathway for iron excretion in humans [12]. The regulation of iron metabolism involves complex mechanisms that balance iron absorption, storage, recycling, and utilization, ensuring the prevention of both iron deficiency and toxicity [13].

The mechanisms of dietary heme absorption are less well understood. In 2005, heme transporter protein 1 (HCP1) was identified in the enterocyte but was subsequently shown to function as a low-affinity heme transporter due to its dual function as a heme/folate importer with high affinity for folate [14,15].

Dietary non-heme iron is predominantly in the oxidized or ferrous (Fe^3+^) form and must be reduced to the Fe^2+^ form, which is the only bioavailable form for intestinal absorption. This reduction is thought to be mediated by ferrireductases in the apical membrane of enterocytes, such as duodenal cytochrome B (DCYTB), facilitated by ascorbic acid [16]. Furthermore, dietary factors such as ascorbic acid chelate ferric iron Fe^3+^ and donate an electron, reducing iron oxidation states to Fe^2+^. Ferrous iron can then be transported by divalent metal transporter 1 (DMT-1), a known mediator of cationic metal transportation located at the apical membrane of the enterocyte [17,18].

Iron taken up by enterocytes can follow three different pathways: It can be used directly for intrinsic cellular metabolic processes, stored, or exported across the basolateral membrane to enter the bloodstream [2].

Iron is stored in the cytoplasm as ferritin, an iron storage protein. To enter the bloodstream, iron passes through ferroportin, an iron transport protein located on the basolateral membrane of enterocytes [19]. Ferroportin is coupled to hephaestin, a ferroxidase that reduces iron to its ferrous form and is located on the enterocytes’ basolateral membranes of the small intestine villi [20,21,22].

Under normal physiological conditions, exported iron is sequestered by plasma transferrin (Tf) (Figure 1). Tf can bind two iron molecules via Tyr (two), His, and Asp residues and a synergistic anion (usually carbonate), maintaining iron in a redox-inert state. Approximately 20–40% of Tf’s iron-binding sites are normally occupied by iron, representing Tf saturation. Tf delivers iron to tissues for uptake by the ubiquitously expressed transferrin receptor 1 (TFR1) composed of a homodimeric protein stabilized by disulfide bonds [23,24].

A clathrin-mediated process is responsible for the endosomal cellular uptake of iron. Endosome acidification by proton pump leads to conformational changes in proteins that result in the release of iron from Tf, which deprived of iron is called apotransferrin (apoTf). Following its reduction by metalloreductase STEAP3, iron moves across the endosomal membrane into the cytoplasm via iron transporter DMT1 [2,25].

Crucially, apoTf and TFR1 complex remain intact at the endosomal pH. Upon return of the endosome to the cell surface, where the pH is approximately 7.4, a conformational change occurs in apoTf leading to its dissociation from TFR1 and recycling of both for subsequent cycles of iron binding and uptake [26,27,28]. In erythroid cells, most of the iron is transported to the mitochondria, where it is incorporated into protoporphyrin for heme synthesis. In non-erythroid cells, iron is stored as ferritin and hemosiderin [29].

Iron metabolism is regulated both at the systemic and cellular levels [30]. At the cellular level, the expression of proteins involved in iron metabolism, such as ferritin or transferrin receptors, is regulated by the interaction of iron-sensitive proteins known as iron regulatory proteins (IRPs) [31]. IRPs are RNA-binding proteins that interact with IRE (iron-responsive elements) to control the translation of iron metabolism proteins. The translation depends on the location of the IRE in the UTR of the target mRNA. When an IRP binds to a single IRE at the 5′ UTR, mRNA translation is repressed. In contrast, the binding of an IRP to an IRE at the 3′ UTR stabilizes the transcript and leads to increased mRNA translation [32]. At the systemic level, iron homeostasis is regulated by the iron-regulating hormone hepcidin. Hepcidin negatively regulates iron absorption. This liver-derived peptide binds to ferroportin on the plasma membrane of enterocytes, macrophages, hepatocytes, and other cells. This binding promotes Jak-dependent phosphorylation and internalization of ferroportin, leading to its lysosomal degradation [33,34].

## 2. The Role of Iron in Colorectal Cancer

Because iron deficiency and surplus influence tumorigenesis, progression, drug resistance, and immunological activation and escape in cancer, it is vitally important to maintain the right balance between these two factors. Significant research has recently been conducted on iron’s relationship to colorectal cancer (CRC). There is a strong association between nutrition and CRC risk, especially for subjects characterized by elevated consumption of heme containing red meat (beef, pork, lamb). In spite of the lack of clarity regarding the exact mechanism, there is increasing evidence linking higher levels of iron to the development and progression of colon cancer [35,36]. Increasing proliferation rates and synthetic/metabolic activities associated with malignancy increase iron demand in cancer cells, enhancing their iron uptake machinery. As aforementioned, the majority of iron in the bloodstream is linked to the iron-transporting protein, Tf. To ensure effective and controlled iron distribution to different tissues, this complex transport process consists of multiple essential parts and pathways.

Iron transport proteins, such as ferritin, hepcidin, and transferrin receptors (TfR1 and TfR2), intricately interact to regulate iron homeostasis. Together, these proteins control the absorption, storage, and release of iron within cells, which affects the amount of iron available for cellular processes [37,38,39]. An important feature shared by various types of cancer cells like CRC is an increase in the number of TfR1 molecules present on their surface. This process enhances the rate of cellular uptake for ferrous ions by the transporter protein. TfR1 has an important role in improving cellular proliferation rates; moreover, it helps maintain adequate levels for ribonucleotide reductase activity because this enzyme needs iron as a cofactor for its viability and function. Therefore, overexpression of these receptors creates an iron-enriched environment, which accelerates the progression of cells through cell cycle phases. Furthermore, both metastasis and genomic instability may be influenced by iron-mediated oxidative stress and hypoxia, which triggers the production of hypoxia-inducible factors (HIFs). The HIF complex consists of two subunits: the oxygen-dependent α-subunit and β-subunit, known as the Aryl Hydrocarbon Nuclear Translocator (ARNT), which is always present in the cell, regardless of oxygen levels [40,41]. Prolyl hydroxylase domain-containing enzymes (PHDs) accelerate the breakdown of HIFα subunits in cells when oxygen levels are sufficient. Those enzymes are 2-oxoglutarate-dependent dioxygenases which require oxygen as a co-substrate and iron binding at their active site. Consequently, PHDs function as a vital link between iron and oxygen levels. PHDs are implicated in the cellular response to oxygen, operating as tumor suppressors by hydroxylating HIFα, causing its destruction under normal oxygen levels. However, their inhibition during hypoxia allows HIFα accumulation and activation of gene transcription. The expression pattern of PHDs in malignancies and their direct relationship with carcinogenesis remain unknown [42,43]. Hydroxylation interacts with the von Hippel-Lindau tumor suppressor protein (pVHL), a component of the VCB ubiquitin E3 ligase complex, resulting in the degradation of HIF-α [44]. Reduced PHD activity during hypoxia permits HIFs-α to accumulate and control genes involved in the adaptive response. Despite being primarily inhibited in hypoxia, PHDs continue to function at low oxygen levels, confirming their sensitivity to oxygen and role as cellular oxygen sensors [45]. The PHD family, which includes PHD1, PHD2, PHD3, and PHD4-TM, shares a common dioxygenase domain [46]. PHD1 and PHD3 regulate HIF-2α, which affects tumor behavior by altering oxygen sensing and glycolytic metabolism. PHDs play a crucial role in tumor formation and influencing angiogenesis. However, their impact on tumor progression varies and is context dependent. PHD3 specifically inhibits colon cancer growth [47,48].

Several studies investigating the onset and progression of colorectal cancer take advantage of the murine model Apc^min/+^. Morin et al. demonstrated in 1997 that patients affected by familial adenomatous polyposis (FAP) syndrome are characterized by mutations of the tumor-suppressor gene *Apc* [49]. Mice called Apc*^min/+^* were developed by Moser et al. in 1990 and the name indicated the heterozygote mutation in the APC gene generating multiple intestinal neoplasia (MIN) [50]. The genetic predisposition of the murine model Apc*^min/+^* is frequently combined with chemical inflammatory stimuli or combining different mutations to evaluate tumor development under various conditions [51,52,53].

Xue et al. used mice with an intestine-specific *Vhl* disruption (VhlΔIE)) crossed with CRC-predisposed mice (Apc*^min/+^*) to reveal that the activation of HIF-2α in VhlΔIE/*Apc^min^*^/+^ mice led to a dramatic increase in colon tumor multiplicity and incidence, with most tumors evolving into carcinomas, while Apc*^min/+^* mice typically develop mainly small intestinal tumors. This activation promotes epithelial cell survival, possibly due to the disruption of iron transport. Knocking down both *Vhl* and *Hif-2α* in VhlΔIE/Apc*^min/+^* mice reduced colon cancer incidence and severity, highlighting the importance of HIF-2α in this process. Analysis of *HIF-2α* target genes indicated that DMT-1, an iron transporter, is overexpressed in colon tumors, contributing to iron buildup and carcinogenesis [54].

The progression of CRC also occurs due to decreased expression of iron export components, mainly HEPH, which causes iron accumulation in the intestines. The transcription factor *CDX2*, which regulates HEPH, is downregulated in poorly differentiated tumors, and exosomal miR-24-3p from cancer-associated fibroblasts inhibits its expression, contributing to chemoresistance. Furthermore, whereas FPN1, another iron exporter, is upregulated in CRC, it is usually inactive and less expressed in the invasive front of tumors. FPN1 appears to be regulated by local hypoxia and increased hepcidin synthesis, implying that modulating *FPN1* expression may affect tumor growth, although translating this into clinical practice is difficult [36].

Cellular adaptation to low oxygen levels is supported by HIFs, which upregulate *TFR1* expression to improve iron absorption. For cell survival, distinction, and multiplication, the Wnt/β-catenin signaling system is relevant. Mutations in *APC*, *CTNNB1* (which encodes β-catenin), and *AXIN* are among the pathway components that cause constitutive activation of β-catenin in CRC. Following this activation, β-catenin is translocated to the nucleus, where it promotes oncogenesis by acting as a transcriptional co-activator for TCF/*LEF* target genes. Based on newly discovered data [55], β-catenin directly stimulates TfR1 transcription, and this transcriptional regulation of *TfR1* by β-catenin establishes a direct relationship between iron metabolism and carcinogenic Wnt signaling.

Ferritin levels have been shown to be elevated in CRC tissues, which may indicate that tumor cells are storing more iron. This overexpression could be a physiological reaction to the higher iron requirements of cancer cells that are multiplying quickly. Iron is necessary for the synthesis of DNA and the creation of energy, two processes that are noticeably elevated in cancer. To ensure an adequate supply of iron, tumor cells consequently frequently upregulate the production of ferritin. Iron builds up inside tumor cells when ferroportin expression is frequently downregulated in CRC. Because the intracellular iron buildup ensures a continuous supply of iron for vital cellular processes, it may encourage the formation of tumors. Moreover, too much iron can catalyze the production of reactive oxygen species, which can damage DNA and promote genetic mutations, both of which can accelerate the development of cancer. Hepcidin expression is frequently changed in CRC, which disturbs iron homeostasis. Iron absorption within cells can be caused by elevated hepcidin levels, which lowers iron availability in the bloodstream. Low hepcidin levels may result in increased iron export and raised serum iron levels, which may induce DNA damage and oxidative stress, both of which may encourage the growth of tumors. Research has demonstrated that hepcidin expression changes in CRC are linked to a poor prognosis and can serve as a biomarker for the advancement of the illness [56]. *HIFs* are frequently activated in the hypoxic tumor microenvironment of CRCs, which leads to the abnormal iron metabolism observed in these tumors. Cancer progresses because this environment is favorable to tumor development and survival owing to the enhanced iron uptake and retention caused by HIFs. Furthermore, oxidative stress and genomic instability might be exacerbated by HIF-induced alterations in iron metabolism, which can further promote cancer.

Recent research discovered a novel path of iron transfer within the TME using Lipocalin-2 (LCN2), a protein previously known for its role in innate immune defense by binding iron-loaded siderophores and limiting pathogen survival [57]. In vitro, experiments have shown that *LCN2* mRNA and protein levels were higher in colon cancer samples than in neighboring normal tissue and altering *LCN2* in CRC cell lines influenced cell proliferation, epithelial-to-mesenchymal transition (EMT), invasion, and metastasis. Furthermore, in vivo tests indicated that *Lcn2* expression affected tumor growth and cellular markers, emphasizing its potential function in controlling CRC phenotypic plasticity, such as E-cadherin and vimentin [36].

A crucial part of iron transport is played by holo-transferrin, also known as transferrin-bound iron. *TFR1*, which is widely expressed in various cell types, and tissue-specific transferrin receptor 2 (*TFR2*) are the genes coding for the proteins with which holo-transferrin interacts. Through a process known as clathrin-mediated endocytosis, these connections aid in the uptake of iron by cells. Iron is eventually released into the cytoplasm for storage or metabolic usage when the transferrin-receptor complex is internalized into the cell during this process inside a clathrin-coated vesicle. Due to their characteristic high iron content, colon cancer cells have been observed to be highly susceptible to ferroptosis, which is a specialized form of programmed cell death controlled by iron-dependent lipid peroxidation and the depletion of glutathione, which can be triggered by diverse stimuli such as oxidative stress and iron overload [58,59]. Ferroptosis occurs when a high level of iron in the cytosol catalyzes the accumulation of lipid peroxides rather than caspase activation and cellular fragmentation as observed during apoptosis. It has been shown that low oxygen levels can alter iron metabolism and ROS production in the body, thereby affecting ferroptosis sensitivity [58,60,61]. During the digestive process in the intestines, iron ions are absorbed by the cells and carried into the bloodstream where they can initiate ferroptosis if necessary. Once inside the cell, iron is managed through a complex system involving TFR1 and the unstable iron pool, with excess iron stored as ferritin to maintain cellular homeostasis [62,63]. Disruption in iron balance can lead to oxidative damage and cell death, as iron catalyzes ROS production through both enzymatic and non-enzymatic lipid reactions. Strategies such as gene knockout of transferrin receptors or enhancing iron storage can mitigate iron overload and inhibit ferroptosis, while iron chelators can prevent oxidative stress by blocking electron transfer from iron. Consequently, regulating iron metabolism and targeting ferritin phagocytosis emerge as promising approaches for controlling ferroptosis and its associated pathologies [63]. Additionally, further investigation of ferroptosis and tumor escape may lead to new and more effective therapies for CRC. Transferrin-bound routes are not the only ones used by the systemic circulation to carry iron. In this context, it is important to underline the results obtained by Pacella et al. demonstrating that CD71 was upregulated on activated Tregs infiltrating human liver cancer, and mice with a Treg-restricted CD71 deficiency fail to correctly expand the Treg repertoire [64]. The axis between iron uptake and Treg proliferation may be crucial for tumor-mediated immune escape. In addition to being present in extracellular environments, non-transferrin bound iron (NTBI) can be absorbed by cells via other pathways unrelated to Tf receptors. As previously described, DMT1 is one important mechanism that permits ferrous iron (Fe^2+^) to enter cells directly. This mechanism is particularly important in conditions where Tf saturation is exceeded, such as in cases of iron overload or certain pathological states. In cancer cells, Tf synthesis supports iron supply and growth through an autocrine mechanism [65,66]. Chronic inflammation is a characteristic feature of CRC, being described as a crucial trigger for CRC in genetically predisposed individuals and murine models of CRC [52,67]. This persistent inflammation can disrupt normal iron metabolism. Inflammatory cytokines, which are signaling molecules involved in inflammation, can modify the expression of proteins that regulate iron within the body. This alteration leads to imbalances in iron homeostasis, which may contribute to the progression of CRC (Figure 2).

## 3. Iron-Rich Diet and Inflammatory Microbiota

### 3.1. Intestinal Microbiota and CRC

Apart from being essential to human physiology, iron also plays a significant role in gut microbiota composition and function [68,69].

The intestine represents the largest surface of the human body in constant contact with the outside world, including foreign antigens, trillions of microbes—collectively named microbiota—and occasionally pathogens. The intestinal epithelium is almost impermeable to bacteria as it is covered with mucus, secreted by enterocytes and calceiform cells. The gut microbiota has been recognized as an important determinant in CRC initiation and progression [70].

Over the years, the intestinal microbiota, influenced by age, diet, use of antibiotics, and other genetic and environmental factors, undergoes an imbalance that leads to a large loss of commensal bacteria and an enrichment of pathogenic bacteria [71]. This dysbiosis contributes to the triggering of pro-inflammatory mechanisms in the intestinal tissue or to the progression of chronic inflammatory diseases, such as ulcerative colitis associated with CRC.

Patients affected by ulcerative colitis present a dysfunction in the production of mucin, a glycoprotein that forms a protective barrier on the intestinal epithelium. Most commensal bacteria live and feed within this layer of mucus. If the production of mucin is inhibited, the epithelium becomes permeable to bacteria that, coming into direct contact with the epithelium, could trigger pro-inflammatory and pro-carcinogenic mechanisms [72].

Recent studies have shed light on the molecular mechanisms that may contribute to the interactions between microbes and the CRC microenvironment.

The inflammatory events that are associated with the development of CRC include DNA damage by reactive oxygen and nitrogen species (ROS and RNS), produced by macrophages, neutrophils, and dendritic cells. Furthermore, as mentioned above, pathogenic bacteria’s invasion of the epithelium triggers an inflammatory process by activating immune cells that secrete and release pro-inflammatory factors such as IL-6, TNF, and cyclooxygenase [73].

In addition to the virulence factors that allow pathogenic bacteria to invade the intestinal mucosa and trigger inflammation, metabolites produced by intestinal microbes are also linked to the progression of CRC. For example, secondary bile acids, produced by the microbiota via biotransformation of primary bile acids, such as deoxycholic acid and lithocholic acid, can directly induce oxidative DNA damage and tumor formation, epithelial-mesenchymal transition, and constitutive activation of the MAPK signaling pathway that contributes to tumor progression and metastasis. In addition, these metabolites also have antimicrobial activity, causing the death of commensal bacteria and thus an increased presence of CRC-related bacteria [74].

Other types of bacteria, instead, play a positive role in the intestinal barrier by producing short-chain fatty acids (SCFA) derived from dietary fiber. The most common SCFA are acetate, propionate, and butyrate [75]. These three secondary metabolites have an anti-inflammatory and anti-oxidative effect and promote mucin production. Butyrate promotes the differentiation of regulatory T cells; the activation of macrophages reduces the secretion of pro-inflammatory cytokines and activates the AP-1 pathway controlling cell proliferation and apoptosis [76].

The diversity of bacteria present in the intestine during carcinogenesis can be used as a biomarker and as a diagnosis for CRC by trying to better understand which bacterial species and which of the metabolites they produce can induce the development and progression of the disease [77].

### 3.2. Intestinal Microbiota Shift in Response to Iron Availability

Numerous host mechanisms that contribute favorably to the growth of commensal bacteria create, at the same time, unfavorable conditions for pathogenic bacteria. Conversely, commensal bacteria also release metabolites and micronutrients that contribute to the healthy homeostasis of the host. Among the most significant metabolites produced by commensals are SCFA derived from bacterial fermentation of dietary fibers in the colonic lumen. Among SCFA, butyrate, propionate, and acetate are better characterized for their role in numerous physiological aspects related to epithelial barrier homeostasis, including favoring the mucosal barrier integrity by promoting the assembly of tight junction proteins. SFCA production results in weakly acidic pH conditions that may contribute to protection from infections and to the absorption of vitamins, electrolytes, and iron. This aspect may be debatable, as lower pH is associated with inflammatory conditions potentially leading to CRC development. Indeed, under healthy conditions, colonic pH gradually increases from 6.0 near the cecum toward around 7.0 in the rectum, and inflammation leads to a pH drop due to immune cell activation, increased energy demand, and glucose consumption via glycolysis [78]. Lumenal pH has been shown to decrease in patients with active ulcerative colitis that is also characterized by intestinal dysbiosis [79]. A fecal transplant from an animal model characterized by chronic intestinal inflammation and dysbiosis into a recipient genetically predisposed to develop CRC confirmed that an inflammatory microbiota represents a sufficient trigger for CRC development in genetically predisposed individuals [80]. Inflammation leads to lower pH in the intestinal lumen; this condition may lead to the dissociation of iron from iron-sequestering proteins released in the intestinal lumen under inflammatory conditions to prevent dysbiosis [81]. Iron availability may create a vicious circle sustaining chronic inflammation and, consequently, CRC development in predisposed patients. This hypothesis requires more data to be confirmed but may represent a different perspective to design adjuvant therapies for CRC prevention and treatment.

Recent studies demonstrated that a low-iron diet is associated with a decrease in relative abundance of SCFA-producing bacteria and consequently a high invasion of pathogenic bacteria into the intestinal lumen. Studies conducted by James R. Ippolito et al. have shown that enterocolitis, induced by *Salmonella Thyphimurium*, was attenuated by an iron-deficient diet. The poor availability of iron in the colon limited the abundance of the pathogen and therefore decreased the severity of the infection compared to an iron-rich environment. This result suggests that iron deficiency negatively affects the composition and function of the intestinal microbiota [82].

A diet rich in iron exerts a significant influence on the composition and function of the intestinal microbiome. In general, part of the iron is absorbed in the duodenum, while the rest passes into the colon where it is available for the gut microbiota. The study of Ke Gu et al., 2023, has demonstrated that iron overload influences the composition of the gut microbiome. For example, *Lactobacillus*, since they grow in an iron-poor environment, give way to other microorganisms in the presence of excess iron. Furthermore, *lactobacillus* and other microorganisms greatly influence intestinal immune functions by modulating inflammation. In fact, in the presence of high iron concentrations a reduction in the abundance of beneficial bacteria, including *Lactobacillus*, *Dubiosella*, and *Bifidobacterium* has been observed. Instead, iron deficiency increases the presence of commensal bacteria and reduces the harmful bacteria [83].

Therefore, a high concentration of iron in the colon, leads to an increase in dysbiosis and consequently the onset of the inflammatory response, an increase of oxidative stress, and mucosal damage. In contrast, low concentrations of iron could reduce inflammation, and oxidative stress in patients with ulcerative colitis. The results from several studies demonstrate that the expression of inflammatory cytokines, such as TNF-alpha and IL-1beta, increases in conditions of iron overload while at low iron concentrations it decreases significantly. This indicates that iron can somehow alter the inflammatory response [83].

These statements can be explained by the following mechanism: At high concentrations iron is mainly absorbed in the small intestine, whereas unabsorbed iron enters the colon, altering the microbiota by reducing the growth of commensal bacteria. This active ferroptosis leads to an inflammatory response. Thus, iron abundance causes colitis. Several studies have also shown that the removal of the intestinal microbiota increases the inflammatory response and promotes ferroptosis; its replenishment attenuates inflammation and inhibits ferroptosis [84].

The mechanism by which these bacteria modulate the inflammatory process is still unclear, but some studies suggest that the release of microbial metabolites can somehow regulate the intestinal inflammatory state. For example, the synthesis of alpha-tocopherol, a metabolite with anti-inflammatory effects that inhibits ferroptosis, is increased in conditions of low iron availability [85].

These data suggest a further mechanism by which bacteria can modulate inflammation: a higher concentration of iron can alter the production of microbial metabolites by dysregulating ferroptosis and triggering inflammatory colitis.

Furthermore, they play a role in bending the immune response towards tolerance. In the intestinal mucosa, intestinal resident immune cells are dynamically educated by the intestinal milieu to become inflammatory impaired and favor homeostasis and eventually tissue repair [86,87]. Physical distance between the intestinal epithelium and the microbial community is granted by the mucus layer, which consists of an inner tight and almost impermeable layer and an outer layer characterized by an expanded volume, which is likely the result of commensal bacterial proteases and glycosidases secretion making it an ideal habitat for the commensal flora [88]. Iron chelators are important players in microbial-host interplay, as iron is a limiting factor for bacterial growth.

Antigen-presenting cells, including macrophages and dendritic cells, in addition to engulfing pathogens, can induce the activation of hepcidin which has the task of sequestering iron making it unavailable for bacterial growth.

Furthermore, the intestine also uses other defense mechanisms that help to grow commensal bacteria and eliminate pathogenic ones, through the presence of neutrophils, the secretion of microbial peptides, immunoglobulins, and cytokines by Paneth cells. No less important are anti-inflammatory cytokines and the activation of regulatory T cells [89].

However, some commensal bacteria, called pathobionts, can be potentially pathogenic in the presence of specific genetic or environmental modifications of the host. This causes a reduction in SCFA synthesis, and an increase in intestinal wall permeability resulting in antigen translocation from the luminal to the basolateral side and, consequently, the production of pro-inflammatory cytokines leading to chronic inflammation, cell and tissue damage. Events perturbing microbial metabolite production or the interaction between intestinal microbiota and enterocytes is typical of some pathologies including inflammatory bowel diseases (IBD) [90].

## 4. Conclusions

Iron is the most abundant trace metal in the human body. Due to the chemical properties of iron, its absorption is restricted to limited portions of the intestine. Once absorption has taken place, numerous strategies are followed to recycle and/or prevent iron loss. As written in Sun Tzu’s *The Art of War*: “A wise general strives to feed off the enemy. Each pound of food taken from the enemy is equivalent to twenty pounds you provide by yourself”. By the same principle, iron intake represents a crucial battlefield between sane and malignant cells. To better respond against potential infections, immune cells—macrophages in particular—uptake iron to starve pathogens and better respond to the metabolic needs of the inflammatory response. The excessive iron intake characteristic of the westernized diet may be responsible for favoring the growth of malignant cells. At the same time, iron abundance in the intestinal lumen may support the switch of the intestinal microbiota towards dysbiosis that in turn supports chronic inflammation, a condition associated with CRC development, particularly in genetically predisposed individuals. The present review aims at highlighting the need for more studies to dissect the axis between iron intake and CRC development.

## Figures and Tables

**Figure 1 ijms-25-12389-f001:**
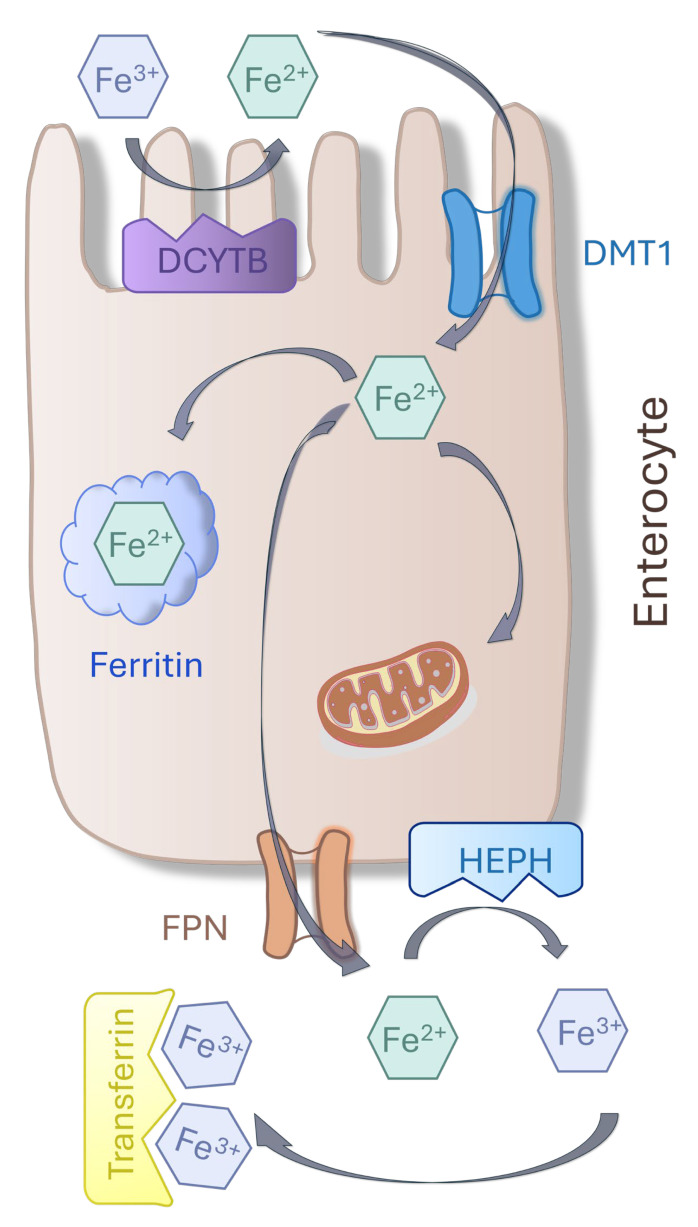
Schematic mechanisms of intestinal iron transport. The primary mechanism by which iron is absorbed into enterocytes is via the divalent metal ion transporter 1 (DMT1) on the luminal membrane after reduction by duodenal cytochrome b (DCYTB). Once in the intracellular labile iron pool, iron can be used directly for intrinsic cellular metabolic processes, stored as ferritin, or exported into circulation via ferroportin (FPN) after reduction by hephaestin (HEPH), a ferroxidase located on the basolateral membrane of enterocytes. Transferrins mediate the transport of iron through blood plasma.

**Figure 2 ijms-25-12389-f002:**
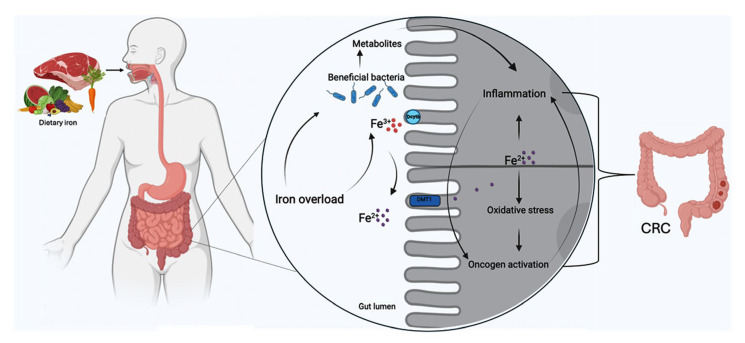
Iron overload and CRC development. Elevated iron uptake by divalent metal transporter 1 (DMT-1) leads to iron accumulation in the colon resulting in several downstream effects. Increased iron levels enhance the production of reactive oxygen species (ROS), which promotes oxidative stress and DNA damage. This oxidative environment supports oncogene activation, further driving tumorigenesis. Additionally, the pro-inflammatory mediators released in response to iron accumulation amplify local inflammation, while changes in the gut microbiome (dysbiosis) further contribute to CRC growth and progression. Together, these processes create a microenvironment conducive to the development and progression of CRC.

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
