# Peer review of "Enhanced CRC Growth in Iron-Rich Environment, Facts and Speculations"

_ijms, 2024, doi:10.3390/ijms252212389_

Round 1

Reviewer 1 Report

Comments and Suggestions for Authors

Chieppa et al. provide valuable insights into iron metabolism; however, the manuscript requires substantial revisions. Key issues include low-quality figures, numerous typographical errors, and awkward sentence structures, which impact the overall readability. The manuscript contains several typos and awkward sentences, and the authors should proofread the text more thoroughly.

Major:

1.    Figures: The quality of the figures is inadequate. Higher-resolution images should be supplied to improve clarity and ensure the data is easily interpreted. This is particularly important for detailed scientific presentations. Additionally, it is unclear which text refers to Figure 1.

2.    Typos and awkward sentences: There are numerous instances of typographical errors and awkward sentence structures. For example:

o    "Iron's ability to participate in redox reactions by donating and accepting electrons makes it essential for various cellular processes..."

This sentence could be revised for clarity and better readability. I recommend a comprehensive review of the manuscript to correct these issues.

3.    Scientific clarity: Although the content is generally relevant, some sections—such as the discussion of iron absorption and transport mechanisms—need clearer explanations. For instance, the role of dietary components in iron bioavailability (e.g., ascorbic acid’s effect on non-heme iron) could be expanded.

Minor:

1.    Line 36: The unit “16 mg/m2” should have a space between the number and unit. This also applies to “6 mg/m2” and “7 mg/m2.”

2.    Line 58: The format for "Fe2+" should be consistent throughout.

3.    Abbreviations such as "Transferrin" and "hypoxia-inducible factors" should be used consistently throughout the manuscript.

4.    Clarify the meaning of “CRC-predisposed mice (Apcmin/+)” and provide more detailed explanations.

5.    Provide abbreviations for "CDX" where appropriate, as there are too many instances to cite individually. 

6. It would be better to show the concentrations of all nutrients, including iron in serum, colon mucosa etc.

Comments on the Quality of English Language

There are numerous awkward sentences and typographical errors. The reviewer strongly recommends that the authors use an English editing service.

Author Response

Dear Editor,

We are very grateful for the reviewers’ constructive comments. We worked to address them all and improve the quality of our review to the standard of International Journal of Molecular Sciences.

Responses to the reviewers’ comments are listed below in red.

REV 1

Major:

  1. Figures: The quality of the figures is inadequate. Higher-resolution images should be supplied to improve clarity and ensure the data is easily interpreted. This is particularly important for detailed scientific presentations. Additionally, it is unclear which text refers to Figure 1.

The quality of the pictures has been improved and should now be acceptable. Figure 1 is no longer mentioned at the end of the paragraph (line 120), but is now in line 68.  

  1. Typos and awkward sentences: There are numerous instances of typographical errors and awkward sentence structures. For example:

o    "Iron's ability to participate in redox reactions by donating and accepting electrons makes it essential for various cellular processes..."

This sentence could be revised for clarity and better readability. I recommend a comprehensive review of the manuscript to correct these issues.

Thanks for your comment, the sentence has been rewritten and should be now more clear.

  1. Scientific clarity: Although the content is generally relevant, some sections—such as the discussion of iron absorption and transport mechanisms—need clearer explanations. For instance, the role of dietary components in iron bioavailability (e.g., ascorbic acid’s effect on non-heme iron) could be expanded.

The manuscript has been corrected following reviewer comments and several sentences were changed to better explain these concepts, in particular from line 52 to line 62

Minor:

  1. Line 36: The unit “16 mg/m2” should have a space between the number and unit. This also applies to “6 mg/m2” and “7 mg/m2.”

The manuscript has been corrected following reviewer comments

  1. Line 58: The format for "Fe2+" should be consistent throughout.

The manuscript has been corrected following reviewer comments

  1. Abbreviations such as "Transferrin" and "hypoxia-inducible factors" should be used consistently throughout the manuscript.

The manuscript has been corrected following reviewer comments

  1. Clarify the meaning of “CRC-predisposed mice (Apcmin/+)” and provide more detailed explanations.

We thank the reviewer for this suggestion, we had a short history about APCmin/+ discovery and references for readers eventually interested in knowing more about this important murine model. The new version of the manuscript includes a description of the APC model from line 180 to 191

  1. Provide abbreviations for "CDX" where appropriate, as there are too many instances to cite individually

We specified CDX in line 203

  1. It would be better to show the concentrations of all nutrients, including iron in serum, colon mucosa etc.

We thank the reviewer for this suggestion. We changed the sentence and we now included the total content of trace metals in the human body line 36-39

Reviewer 2 Report

Comments and Suggestions for Authors

Manuscript ijms-3269045

„Enhanced CRC growth in iron-rich environment, facts and speculations” for International Journal of Molecular Sciences

Comments:

The manuscript is very interesting, presenting important facts concerning alimentary tract cancer, specifically colon cancer and iron.

1.      I have only one proposal for manuscript supplementation. Please expand text fragment concerning ferroptosis and its impact on colon cancer development.

2.      Please also extract clear conclusion section based on the data presented in the manuscript text.

Author Response

Dear Editor,

We are very grateful for the reviewers’ constructive comments. We worked to address them all and improve the quality of our review to the standard of International Journal of Molecular Sciences.

Responses to the reviewers’ comments are listed below in red.

REV 2

-Comments:

The manuscript is very interesting, presenting important facts concerning alimentary tract cancer, specifically colon cancer and iron.

  1. I have only one proposal for manuscript supplementation. Please expand text fragment concerning ferroptosis and its impact on colon cancer development.

We thank the reviewer for the positive comments and constructive suggestions. The new version of the manuscript includes a section dedicated to ferroptosis from line 258 - 278

  1. Please also extract clear conclusion section based on the data presented in the manuscript text.

The new version of the manuscript includes a brief conclusion paragraph.

Round 2

Reviewer 1 Report

Comments and Suggestions for Authors

The authors have not improved the figures. The quality of the figures remains suboptimal, and they need to be more detailed and informative.

Additionally, I recommend that the authors use an English editing service to address typos issues throughout the manuscript.

Comments on the Quality of English Language

Reviewer highly recommend using an English editing service since there are still too many mistype. 

Author Response

We thank the reviewer for these comments, we hope that in the present version the manuscript will be considered acceptable.

The authors have not improved the figures. The quality of the figures remains suboptimal, and they need to be more detailed and informative.

We have changed figure 1 and its position in the text. The quality of the figure was 300dpi in the past version (uploaded as separate PDF file in the previous version). We hope that this will be considered more informative and we are ready to improve the digital quality if needed.

Additionally, I recommend that the authors use an English editing service to address typos issues throughout the manuscript.

The manuscript has been revised by Dr. Rosalind Polley, PhD born and graduated in London. She was a post doctoral fellow in my same laboratory of Immunology at the NIH in Bethesda, MD, USA. I trust her English revision will improve our previous version.

Round 3

Reviewer 1 Report

Comments and Suggestions for Authors

Please use an English editing service, as there are still numerous awkward sentences.

Comments on the Quality of English Language

Please use an English editing service, as there are still numerous awkward sentences